# Molecular Basis and Rationale for the Use of Targeted Agents and Immunotherapy in Sinonasal Cancers

**DOI:** 10.3390/jcm11226787

**Published:** 2022-11-16

**Authors:** Andrea Esposito, Erika Stucchi, Maria Baronchelli, Pierluigi Di Mauro, Marco Ferrari, Luigi Lorini, Cristina Gurizzan, Nyall Robert Jr London, Mario Hermsen, Matt Lechner, Paolo Bossi

**Affiliations:** 1Medical Oncology Unit, Department of Medical and Surgical Specialities, Radiological Sciences and Public Health University of Brescia, ASST-Spedali Civili, 25123 Brescia, Italy; 2Section of Otorhinolaryngology, Head and Neck Surgery, Department of Neurosciences, Azienda Ospedaliera of Padua, University of Padua, 35128 Padua, Italy; 3Head & Neck Surgery, Department of Otorhinolaryngology, Johns Hopkins School of Medicine, Baltimore, MD 21205, USA; 4Department of Head and Neck Oncology, Instituto de Investigaciòn Sanitaria del Principado de Asturia, 33011 Oviedo, Spain; 5UCL Cancer Institute, University College London, London WC1E 6BT, UK; 6Division of Surgery and Interventional Science, Academic Head and Neck Centre University College London, London WC1E 6BT, UK

**Keywords:** molecular, immunotherapy, target therapy, sinonasal cancers, biomarkers

## Abstract

Despite the progress of surgery, radiotherapy, and neoadjuvant chemotherapy, the prognosis for advanced sinonasal cancers (SNCs) remains poor. In the era of precision medicine, more research has been conducted on the molecular pathways and recurrent mutations of SNCs, with the aim of understanding carcinogenesis, helping with diagnosis, identifying prognostic factors, and finding potentially targetable mutations. In the treatment of SNC, immunotherapy is rarely used, and no targeted therapies have been approved, partly because these tumors are usually excluded from major clinical trials. Data on the efficacy of targeted agents and immune checkpoint inhibitors are scarce. Despite those issues, a tumor-agnostic treatment approach based on targeted drugs against a detected genetic mutation is growing in several settings and cancer subtypes, and could also be proposed for SNCs. Our work aims to provide an overview of the main molecular pathways altered in the different epithelial subtypes of sinonasal and skull base tumors, focusing on the possible actionable mutations for which potential target therapies are already approved in other cancer types.

## 1. Introduction

Sinonasal cancers (SNCs) include different tumors of the nasal cavities, maxillary, sphenoidal, ethmoidal, and frontal sinuses. They are rare, with an annual incidence of approximately one case per 100,000 inhabitants worldwide [1]. In Italy, more than 300 SNCs are registered every year [2].

Epithelial SNCs include different histological subtypes: the most common is squamous cell carcinoma (SCC), either keratinizing or non-keratinizing, followed by adenocarcinoma (intestinal-type or non-intestinal type), sinonasal undifferentiated carcinoma (SNUC), sinonasal neuroendocrine carcinoma (SNEC), NUT carcinoma, lymphoepithelial carcinoma, teratocarcinosarcoma, and minor salivary gland tumors.

There are difficulties in diagnosing and treating these tumors because oftheir rarity, histological diversity, and proximity to vital structures (such as orbit, skull base, and brain). The standard of care for advanced SNCs remains a multimodal approach based on radical surgery followed by adjuvant radiotherapy. Advances in imaging techniques, surgical and endoscopic approaches [3], radiotherapy modalities [4] (intensity-modulated radiation therapy, volumetric modulated arc therapy and heavy-ion radiations) and neoadjuvant chemotherapy-containing strategies have shown promising results in improving outcomes. However, the prognosis remains poor [3].

No targeted therapies or immunotherapies are approved for SNCs. These drugs are uncommonly administered in curative and palliative settings unless it’s in clinical trials or expanded biomarker-based use. Data on their use are limited and usually derived from case reports. Recently, efforts have been made to perform gene sequencing on SNCs and to find actionable target mutations. Different studies have shown the potential advantages and clinical implications of this approach [5,6]. Basket trials may represent a useful tool for tumor-agnostic drug development, but no definitive outcomes have been published for SNCs.

The present work aims to summarize the main potentially actionable altered molecular pathways for each epithelial subtype of SNC and to critically examine the data on the clinical application of target drugs/immunotherapies in SNCs. We performed an overview of the literature via PubMed, analyzing those English-written papers in which data on genomic alterations and chromosomal aberrations of each epithelial SNCs were reported. Single-case reports were also included. The research was conducted till July 2022, with the following keywords: “sinonasal cancer”, “sinonasal squamous cell carcinoma”, “sinonasal intestinal-type adenocarcinoma”, “sinonasal non intestinal-type adenocarcinoma, “sinonasal neuroendocrine carcinoma”, “sinonasal undifferentiated carcinoma”, “NUT carcinoma”, “teratocarcinosarcoma”, “sinonasal lymphoepithelial carcinoma”, “molecular alterations”, “targeted therapies”, and “immune biomarkers”. Then, we analyzed if the reported alterations were actionable using two different tools: OncoKb (OncoKB™—Memorial Sloan-Kettering’s Precision Oncology Knowledge Base) and My Cancer Genome (www.mycancergenome.org, accessed on 31 July 2022). We focused on approved drugs by at least one regulatory agency until July 2022.

## 2. Squamous Cell Carcinoma

Sinonasal SCC (SNSCC) is the most common histological subtype (60–75%) of the skull base, with an incidence of 35–58% [7] and a 5-year mortality rate of ~40% [8]. 

The genetic characterization of SNSCC is showing promising results. In 2015, Udager et al. [9] analyzed the presence of pathogenic somatic mutations in SNSCC. This study showed a high prevalence of EGFR alterations (88%) in the inverted papillomas (IPs) and IP-associated SNSCC cases (77%). In contrast, no EGFR alterations were observed in the non-IP-associated SNSCC and in other papillomas. The most common EGFR alteration identified was exon 20-insertion (ins), involving residues located between A767 and V774. Other less common EGFR alterations were deletion-insertion in exon 19 and single nucleotide substitution in exon 19. In addition, in de novo SNSCC EGFR gene amplifications have been documented in about 30% of cases [9,10].

Since several therapies are approved for treating EGFR-mutant non small-cell lung cancer (NSCLC), unique treatment opportunities may open. The potential utility of first-generation EGFR-inhibitors (gefitinib and erlotinib) and second-generation EGFR-inhibitors (neratinib, afatinib, dacomitinib) in the context of SNSCC has been investigated and has shown limited results [9]. This could be explained by the high prevalence of EGFR exon 20-ins, which are resistant to these drugs [9], but are more susceptible to new target therapies, including amivantamab [11] and mobocertinib [12], recently studied and approved in NSCLC. Trials with these molecules in SNSCC are desirable. Even poziotinib (HM781–36B), an irreversible EGFR inhibitor, has been studied in different clinical trials, showing efficacy in NSCLC [13] and in recurrent and/or metastatic head and neck SCC (R/M-HNSCC) [14]. 

ERBB2 copy number gain is another genetic alteration found in SNSCC with an incidence of 21% and elevated protein expression levels of 7%. The ERBB2 amplification and overexpression correlated with higher tumor stage (T4), intracranial dissemination, and worse outcomes [15]. Several agents have been to treatHER2-overexpressing breast cancer and metastatic gastric or gastroesophageal junction adenocarcinoma. The efficacy of anti-HER2 agents might also be tested in SNSCC.

Schrock et al. [16] examined the FGFR1 gene copy number status in patients with SNSCC. FGFR1 amplification was found in 20% of the SNSCC and 33% of the IP-associated carcinomas. The FGFR1 amplification could represent a potential molecular target for specific FGFR1 inhibitors therapy. Studies have shown that inhibition of FGFR1 significantly reduces tumor cell numbers in FGFR1-amplified NSCLC [17]. In addition, erdafitinib [18] and pemigatinib [19] are being studied in phase I basket trials showing promising results. Both of these drugs are approved for FGFR2 fusion. However, no specific drug for FGFR1 amplifications has been approved at present. 

Muñoz-Cordero et al. [20] found the loss of the PTEN expression in 61% of cases, the overexpression of the AKT in 35% and the overexpression of mTOR in 15% of SNSCC. Those alterations lead to the activation of the PI3K/AKT/mTOR pathway, which could be a potentially actionable target. However, no specific drugs are actually approved for AKT and mTOR overexpression or for PTEN loss. Still two AKT inhibitor drugs, capivasertib and ipatasertib, are investigated in metastatic triple-negative breast cancer [21,22] and metastatic prostate cancer [23,24]. In addition, a pre-clinical study with capivasertib/saracatinib (anti-SRC) for HNSCC was published with promising results [25]. Currently, no trials on these two molecules enroll patients with SNC.

Over the years, other SNSCC mutations have been detected, but none of these can be used as therapeutic targets. Udager et al. [26] identified KRAS mutation in 100% of the oncocytic papillomas (OPs) and 100% of the OP-associated SNSCCs (4–17% of SNSCC) [27]. In particular, the main KRAS mutations detected in the OP-associated SNSCCs were the G12V (60%) and G12D (40%). In contrast, KRAS mutations were found in 5% of the SNSCC without known previous papilloma and in 77% of IP-associated SNSCC [28]. Novel target therapies are under investigation for targeting KRAS (e.g., sotorasib and adagrasib in KRAS G12C mutation), but no drug is currently approved for other common alterations [29]. 

Further non-actionable genomic alterations were found in SNSCC. A higher frequency of p53 expression in SNSCC was reported by several studies, ranging from 33.3% to 100% [30]. Brown et al. [29] identified CDKN2A inactivation in 72.4% of the SNSCC, through mutation and subsequent loss of heterozygosity or focal ‘deep deletion’ of the gene locus. At the same time, it was not detected in sinonasal papillomas. Overexpression of TrkB [31] was identified in 70.4% of SNSCC analyzed and was associated with poor prognosis. SOX2 amplifications were identified in 35% of SNSCC [32]. Other SNSCC minor molecular alterations are TERT copy number gains (27.6%) without TERT promoter mutations, NFE2L2 mutation, CCND1 and MYC copy number gain [29] and CARD11 mutation [32]. Finally, 3.2% of sinonasal tumors showed a deficiency of mismatch repair proteins and/or high microsatellite instability (dMMR/MSI-H), which may confer clinical benefit to immune checkpoint inhibitors (ICIs) treatment [33]. 

To conclude, the DEK::AFF2 fusion-associated carcinoma was recently detected as a distinct variant of SNSCC [34,35]. In a patient with DEK::AFF2 fusion-associated carcinoma, an exceptional response to ICIs was identified [34,36]. 

In addition, HPV-related sinonasal carcinoma could be considered a distinctive carcinoma type [27]. In 2020, Svajdler et al. found transcriptionally active HPV infections in 25% of the SNSCC studied, and confirmed the cancerogenic role of HPV infection in these tumors [37]. In several studies HPV-related SNSCC showed a favorable prognosis and better overall survival (OS) and disease-free survival [27]. 

Actionable genetic alterations of SNSCC are summarized in Table 1.

## 3. Intestinal-Type Adenocarcinoma (ITAC)

Intestinal-type adenocarcinoma (ITAC) is the most frequent adenocarcinoma of the skull base and occurs predominantly in the ethmoid sinuses (40–85%) [38,39]. Franchi et al. [40] suggested that ITAC could arise from premalignant intestinal metaplasia of respiratory and/or glandular epithelium.

ITACs are named for their histologic resemblance to adenocarcinoma of the intestinal tract. Since the late 1990s and early 2000s, researchers have considered ITAC and colorectal adenocarcinoma molecular pathways to overlap in different studies [41,42]. 

In recent years, authors pioneered different gene expression profiling studies of ITAC to better understand the molecular events involved in carcinogenesis and to identify potentially novel markers. The heterogeneous mutational profile of ITAC comprises alterations in different genes.

TP53 is the most frequently mutated gene (40–50%, up to 86%) and no target drugs are available. However, there are ongoing trials on the potential role of WEE1 inhibitors (such as adavosertib [43]) in p53-mutated or deficient cancer cells. p53 status may be used to predict response to chemotherapy [44,45].

KRAS and HRAS mutations have been found in one of 12 (8%) and in five of 31 (16%) ITACs, respectively [46,47]. The frequency of KRAS mutations in sinonasal carcinomas is lower than the 30–45% reported in colorectal cancer [48]. The KRAS mutations primarily consist of base pair changes in three hotspots, corresponding to codons 12 and 13 in exon 1 and codon 61 in exon 2 [49]. No specific target agents are available for these types of KRAS mutations. Pérez et al. [50] analyzed 31 ITACs for the presence of HRAS mutations: G12V alteration appears to be the most frequent in the HRAS gene (16%). HRAS mutations were related to a worse prognosis. In another study, no HRAS mutations were found [51]. Tipifarnib, a farnesyltransferase inhibitor that disrupts HRAS function, has been investigated in metastatic HNSCC with high mHRAS variant allele frequency, showing promising efficacy [52]. NRAS mutations have been infrequently described in ITAC [53].

EGFR amplifications and/or overexpression are present in a substantial subset of ITACs with a colonic differentiation pattern [54]. EGFR gene copy number gains occur in 38–55% of the cases, mostly in the context of a whole chromosome 7 gain. High-level amplification is reported to be rather infrequent, between 2% and 16%. The frequency of EGFR alterations observed in ITAC is lower than in colorectal cancer, lung cancer, or HNSCC [55] and SNSCC. EGFR overexpressed ITAC could be potentially treated with EGFR inhibitors.

Most ITACs carry genetic alterations in four different pathways: Wnt/b catenin, DNA damage response (ATM, BRCA 1 and 2), MAPK and PI3K pathways. This means many ITACs might be treated with specific inhibitors of these pathways. Promising specific therapies targeting the Wnt pathway are currently under investigation in phase I clinical trials [56,57]. Treatment with PARP inhibitors may be considered for ATM, BRCA1 or BRCA2-mutant ITACs. PIK3CA mutations may be susceptible to PIK3CA inhibitors (alpelisib), mTOR inhibitors or new molecules such as AKT inhibitors (capivartesib and ipatasertib). 

Although emphasis is placed on these four signaling pathways, other potentially actionable mutations have been found. BRAF mutations have been rarely seen in a subset of ITAC. MET inhibitors represent another interesting treatment option since MET-activating mutation can be found in up to 64% of ITACs 49; other possible opportunities could be trametinib or cobimetinib in NF1-mutated, anti-HER2 (such as trastuzumab, trastuzumab-deruxtecan and trastuzumab-emtansine) in ERBB2-mutated [58], anti-IDH1 in IDH1- mutated ITAC [53]. However, at the moment, no efficacy data are present in the literature about targeted agents agnostically used in ITAC treatment. Results are shown in Table 2.

## 4. Non-Intestinal Type Adenocarcinomas (N-ITAC)

Sinonasal non-intestinal type adenocarcinoma (N-ITAC) is an extremely rare adenocarcinoma, which morphologically presents neither intestinal-type nor salivary-type adenocarcinoma aspects [59]. According to immunohistochemistry, this type of tumor shows respiratory-type features.

Different variants of N-ITAC are commonly divided into two categories: low grade (with a particular subset of seromucinous adenocarcinoma) and high grade (blastomatous, oncocytic/mucinous, apocrine, poorly differentiated and undifferentiated types) [60]. There is also a very rare distinct form of N-ITAC, the renal cell-like adenocarcinoma [61]. Differences between these histological subtypes are related to the expression of different biomarkers detected using IHC [62]. 

Few studies have analyzed the mutational landscape of N-ITAC. Yom et al. [49] noted a small subset of N-ITAC cases showing p53 overexpression, whereas other cases did not show any genetic abnormalities in KRAS, APC, CTNNB1, DNA mismatch repair genes, or TP53. Another study by Franchi et al. [63] reported that two cases contained a BRAF V600E mutation detected by direct sequencing. BRAF inhibitors may be a therapeutic option for a small quote of N-ITAC with EGFR overexpression and BRAF mutations.

Furthermore, in two studies, Andreasen et al. described three low-grade non-ITAC cases showing ETV6 gene rearrangements, including two cases with ETV6-NTRK3 fusion [64] and one with ETV6-RET fusion [65]. ETV6-rearranged low-grade sinonasal adenocarcinomas can be considered morphologically distinct entities [66]. NTRK inhibitors (such as larotrectinib and entrectinib) may be a therapeutic option for N-ITAC with ETV6-NTRK3 fusion, while anti-RET (selpercatinib and pralsetinib) may be a possible therapeutic option for N-ITAC with ETV6-RET fusion. With further molecular investigations, other tumors falling into the category of N-ITAC will likely be separated into more specific entities.

Results are summarized in Table 3.

## 5. Sinonasal Neuroendocrine Carcinoma (SNEC) 

SNECsare rare poorly differentiated carcinoma with neuroendocrine differentiation, characterized by poor prognosis and a high tendency to relapse. According to the new WHO classification, the diagnostic term of neuroendocrine carcinoma can be applied only to poorly differentiated epithelial neuroendocrine neoplasms [67]. Actually, the SNEC standard of care management is represented by the combination of surgical resection, systemic chemotherapy and radiation therapy. However, the treatment efficacy remains sub-optimal; therefore, the molecular landscape should be explored to increase survival rates by discovering new potential therapeutic targets [68].

Mutations of IDH2 have been identified in SNEC. Different studies (Riobello et al. [69], Gloss et al. [70] and Dogan et al. [71]), analyzing a cohort of several IDH2-mutated sinonasal tumor samples, showed that 11% (1/9), 20.5% (8/39), and 83% (5/6) was diagnosed as SNEC, respectively. Gloss et al. [70] evaluated the frequency of IDH2 variants (*n* = 27), of which the most frequent were R172S (70.4%), followed by R172T (14.8%), R172G (11.1%) and lastly R172M (3.7%). IDH2 mutations represent a possible therapeutic target: enasidenib, an anti-IDH2 agent, has recently been approved by FDA for patients with relapsed or refractory acute myeloid leukemia [72].

IDH2 wild-type SNECs are characterized by ARID1A mutations [71], TP53 mutations (33%, 3/9), and (56%, 5/9) alterations in Wnt pathway genes including CTNNB1 (33%, 3/9), AMER1 (22%, 2/9) and APC (11%, 1/9). Among these mutations mentioned, since ARID1A is a subunit of the SWI/SNF chromatin-remodelling complex, it may be a potential target of EZH2 inhibitors [73].

SMARCB1-deficient carcinomas have also been described among SNEC. They represent an aggressive and poor-prognosis subgroup of sinonasal tumors, characterized by INI1 loss mostly due to homozygous SMARCB1 deletion [74]. SMARCB1/INI-1 (also known as BAF47) is a core subunit of the SWI/SNF complex, and acts as a tumor suppressor by regulating gene transcription and cell proliferation. SWI/SNF tumor suppressor proteins act as antagonists of the polycomb enhancer gene of zeste homolog 2 (EZH2), whereby the EZH2 oncogene is constitutively activated in INI-1-deficient tumors and regulates histone methylation resulting in tumor-suppressor gene silencing, oncogenic transformation, metastasis development, and drug resistance [75,76,77]. Recently, in a phase II basket trial [78], a selective inhibitor of EZH2, tazemetostat, showed clinical activity in patients with advanced epithelioid sarcoma with loss of INI-1/SMARCB1.

Results are shown in Table 4.

## 6. Sinonasal Undifferentiated Carcinomas (SNUC)

Sinonasal Undifferentiated Carcinomas (SNUC) are highly aggressive epithelial tumors with uncertain histogenesis, lacking squamous or glandular differentiation; diagnosis is often challenging and is usually made by exclusion [79]. Because of their aggressive clinical behaviour, they are usually diagnosed as locally advanced, mainly from dural and/or orbital invasion [80,81]. Owing to their chemosensitivity the standard approach is based on neoadjuvant chemotherapy followed by either chemoradiation or surgery followed by postoperative radiotherapy [82]. However, the prognosis remains poor, with a median OS of 22 months [83].

IDH2 mutations are the most frequent genetic alterations in SNUC. The positivity of IDH2 11C8B1 on IHC in sinonasal carcinomas would be highly predictive of the presence of IDH2 R172S/T mutations in around 70% of cases [84]. In a study [71], 88% (14 of 16) of SNUCs hadIDH2 R172X mutations, a global methylation phenotype. and an increase in repressive trimethylation of H3K27. These epigenetic alterations severely reduce gene expression, thus preventing cellular differentiation [85]. In another study [86], authors performed an NGS on 11 cases of SNUCs, identifying IDH2 R172X mutations in 55% of cases, R172S, R172T, and R172M. Several concomitant oncogenic alterations, such as PIK3CA, mTOR, SOX2, and SOX9 were also identified. Using both IHC and NGS, other authors [68] demonstrated the presence of mutations in IDH2 in SNUCs with 11/36 (31%) cases affected, with R172S and R172G as sequence variants. The most important copy number alterations in the IDH2-mutated tumors were gains on chromosome arm 1q and combined loss of 17p and gain of 17q and loss of 22q. To note, these IDH2 mutations act both as positive prognostic and potentially predictive biomarkers. IDH2 is an interesting potential target for IDH inhibitors [72].

A reduced/loss of SMARCB1 expression has also been documented in SNUC [87,88,89,90]. Saleh et al. [91] reported a case of a 45-year-old patient with a locoregional relapse 14 years after diagnosis of an advanced SNUC treated with tazemetostat. The patient received tazemetostat 800 mg twice daily (after first-line etoposide-carboplatin) by maintaining a stable disease for 13 months.

More studies have been conducted on the molecular landscape of SNUC in recent years. Chernock et al. [92] identified the expression of EGFR, c-KIT (CD117), and HER2/neu in SNUC. By IHC, nine of 11 cases (81.8%) were diffusely positive for c-KIT (the samples were analyzed by PCR with appropriate c-kit exon 9, 11, 13, or 17 primers), three of 11 cases (27.3%) were positive for EGFR, and none of the cases were positive for HER2/neu. Neither activating mutation nor gene amplification of c-KIT was detected in these analyzed cases. The lack of activating mutations in c-KIT was confirmed in another study [93], thus limiting the possibility of tackling c-KIT overexpressing SNUCs with targeted agents.

A preclinical study [94] suggests that conventional HER2 immunohistochemical staining is not the best way to investigate the status of HER2 in SNUC specimens, showing a negative result for HER2 staining by IHC versus a strong expression with Western blotting. Since several anti-HER2 treatments are approved for other cancers, authors demonstrated the activity of Lapatinib and Trastuzumab in cell lines and animal models. In the absence of SNUC included in anti-HER2 basket trials, this opportunity deserves to be studied in such trials.

Results are summarized in Table 5.

## 7. NUT Carcinoma (NC)

NC is a rare and aggressive subtype of poorly differentiated squamous carcinoma, genetically defined by the rearrangement of the NUT (recently renamed NUTM1) gene. In approximately 70% of cases, NUTM1 is involved in a balanced translocation with the BET family gene BRD4 on chromosome 19p13.1 [t (15; 19) (q14; p13.1)], forming the BRD4-NUT fusion oncogene. In the remaining 30% of cases, the NUTM1 gene is fused with BRD3 (25%) on chromosome 9 [t (9; 15) (q34.2; q14)], the histone methyltransferase NSD3 on chromosome 8 [t (8; 15) (p11.23; q14)] or ZNF532 on chromosome 18 [t (15; 18) (q14; q23)] [95]. The outcome of the patients with NC is often dismal, with a median survival of only 6.7 months [96]. Unfortunately, all the chemotherapeutic agents tested, including doxorubicin-based regimens, have not shown improved outcomes [97]. Based on these data, there is a clear need to find new therapeutic strategies for this aggressive cancer. Recently, several studies evaluated the efficacy of the BET inhibitors (BETi), drugs with acetyl-histone mimetics compounds that target BRD4-NUT by competitively inhibiting its binding to chromatin. The first proof of the clinical activity of a BETi in NCwas presented by OTX015/MK-8628 [95,98,99]. Other phase I trials are currently evaluating BETi in NC [95], like Birabresib [100] and Molibresib [101]. Despite these promising results, not all patients with NC respond to the BETi. Liao et al. [102] identified six potential pathways that could mediate treatment resistance to BET inhibitors, like MYC and MYC-related genes, RTK and GPCR/cAMP/PKA signaling pathway, TGF-β, Kruppel-like factor 4 (KLF4) and cyclin D1/3. In particular, the cyclin-dependent kinase 4/6 inhibitors appear to have a synergistic effect with BETis on NC, suggesting the rationale for combining therapies in NC [102].

The histone deacetylase inhibitors (HDACi) represent another therapeutic approach for NC. Schwartz et al. [103] identified that the expression of BRD4-NUT is associated with globally decreased histone acetylation and transcriptional repression, which could be restored by treating the NC with histone deacetylase inhibitors (HDACi). A child was treated with the histone deacetylase inhibitors Vorinostat, showing an objective response after 5 weeks of therapy [104]. Also, Maher et al. [105] presented a case of metastatic NC with a partial response to Vorinostat. Based on this evidence, other histone deacetylase inhibitors, like Romidepsin [106] and Belinostat [107], could be considered for the NC treatment. Currently, there is an ongoing phase I trial for CUDC-907, an orally bioavailable HDAC and PI3K inhibitor, in patients with NC and a clinical trial for patients affected by NC resistant to bromodomain inhibitors (NCT02307240).

## 8. Teratocarcinosarcoma (TCS)

TCS are aggressive tumors arising primarily in the sinonasal area and anterior cranial base. They are extremely rare, with less than 100 cases ever reported in the literature. They have different features of malignant teratoma, epithelial cells, neural cells, and mesenchymal elements [108]. Little information is available on TCS biology and tumorigenesis and few clinical data can be derived from case reports.

Rooper et al. found a loss of SMARCA4 expression in 18 cases of 22 sinonasal TCS (82%) and variable positivity for Claudin-4 [109]. Complete loss of SMARCA4 expression in 68% of TCS by IHC, with NGS confirmation of biallelic SMARCA4 inactivation in three cases. These results provide important information about the emerging role of SMARCA4 in SNCs. They particularly suggest that TCS is on a spectrum with SMARCA4-deficient sinonasal carcinomas which show overlapping morphology and molecular characteristics, further readjusting the classification of high-grade sinonasal tumors [108]. In SMARCA4-loss ovarian cancer cells [110], tazemetostat (EZH2 inhibitor) showed a potential benefit. There are also in vitro and in vivo data on susceptibility to CDK4/6 inhibitors and ICIs in SMARCA4-loss ovarian cancer [111].

In a case report, authors [112] found the presence of the p.H1047L activating mutation in the PIK3CA gene, suggesting a potential driving role of the PI3K/AKT/mTOR pathway in tumorigenesis. In the same patient, authors also found a germline alteration in the DDR2 gene (p. Pro476Leu) whose oncogenic function is still considered unknown. The potential involvement of Wnt/β-catenin and PI3K/AKT/mTOR pathways could lead to the application of target therapies for this tumor.

## 9. Sinonasal Lymphoepithelial Carcinoma (SLEC)

Lymphoepithelial carcinoma (LEC) was described for the first time in literature by Schminke [113] and Regaud [114] in 1921. Sinonasal lymphoepithelial carcinoma (SLEC) is an extremely rare neoplasm with approximately 40 cases recognized in the literature. It can be considered an SCC morphologically similar to nonkeratinizing nasopharyngeal carcinoma, an undifferentiated subtype, with a reactive intermixed lymphoplasmacytic infiltrate [115].

There are no data in the literature on altered molecular pathways in this very rare type of sinonasal tumor and there is no evidence of potential molecular targets. However, the neoplastic microenvironment is characterized by an important nonneoplastic lymphoplasmacytic infiltrate cells (including CD8^+^T lymphocytes) between and around tumor nests and high expression of PD-1/PD-L1. Even though data from studies on LEC of other head and neck sites show that MSI and loss of expression of the DNA mismatch repair proteins are not common, there is a potential role for immunotherapy in SLEC [116,117].

## 10. Immune-Check Point Inhibitors: Rationale and Clinical Applications 

ICIs are now the standard of care alone or in combination with chemotherapy in PD-L1 positive recurrent and/or metastatic HNSCC. Although this is still subject to further investigations, predictors of the clinical efficacy of ICIs appear to include high membranous PD-L1 levels of expression, high tumor mutational burden, mismatch repair proteins deficiency, microsatellite instability, and infiltrating leukocyte cells. However, the prognostic and therapeutic role of these biomarkers in SNCs is still poorly known and data on the efficacy of immunotherapy in SNCs are lacking. A summary of the published studies is presented in Table 6.

### 10.1. Immuno-Markers in Sinonasal Cancers

#### 10.1.1. Deficient Mismatch Repair Proteins (d-MMR) and Microsatellite Instability (MSI)

Only a few studies have addressed the MSI/MMR status in sinonasal carcinomas, with a resulting frequency of MSI for ITACs of 2% [118] and between 2–21% in d-MMR/MSI for SNSCCs [119,120,121] 

In a study [33], authors analyzed the presence of d-MMR/MSI sinonasal tumors by testing MMR protein expression using immunohistochemistry (IHC) in 174 SNCs’ samples, including SNSCC, adenocarcinoma, SNEC, and SNUC. Only SNSCC were characterized by the presence of d-MMR/MSI with a frequency of 3.2% (4/125), while all analyzed sinonasal adenocarcinoma types as well as SNUC and SNEC displayed intact MMR protein expression patterns. Although d-MMR/MSI SNSCCs are a small subgroup of SNSCC, they are clearly molecularly defined and may be most likely sensitive to ICIs. 

In a recent study [120] just three of 131 (2.3%) SNSCC showed d-MMR expression, whereas the other 128 (97.7%) cases showed intact expression of all four MMR proteins. All three d-MMR cases showed concurrent loss of MLH1 and PMS2 expression. The authors also tried to analyze themutual relationship with other cancer and/or subject characteristics. In particular, these three tumors did not have a synchronous or metachronous inverted sinonasal papilloma component, nor did they display HPV positivity, EGFR mutation, and EGFR copy number gain.

#### 10.1.2. PD-L1 Expression

Riobello et al. [122] analyzed the expression of PD-L1 in 53 SNSCC and 126 ITAC samples. Membranous PD-L1 staining in at least 5% of tumor cells was observed in 34% (18/53) of SNSCC and 17% (22/126) of ITAC. Expression in >50% of tumor cells was frequent in SNSCC (14/53; 26%) in contrast to ITAC (4/126; 3%). Surprisingly, the nuclear expression of PD-L1 was exclusively observed in papillary/colonic-type ITAC; both SNSCC and ITAC with >5% PD-L1 expression had significantly worse disease-free survival, when treated with standard therapeutic options.

Quan et al. [125] evaluated PD-L1 expression in 96 SNSCC cases. Membranous PD-L1 expression in >5% of tumor cells was observed in 29 patients (30.2%). PD-L1-positive SNSCC cases tended to have a higher lymph node metastasis rate than PD-L1—negative SNSCC (20.7% vs. 10.4%). The Chinese group also found that PD-L1 expression was strongly associated with CD8^+^ and Foxp3+ T-cell infiltration levels in SNSCC, indicating that the PD-1/PD-L1 pathway might be a promising target.

In a recent analysis [120] of a total of 131 SNSCC, 60 showed PD-L1 expression in ≥1% (tumor proportion score, TPS). The TPSs were subdivided into low (1−19%; n = 43, 32.8%), high (20−49%; n = 12, 9.2%) and very high (≥50%; n = 5, 3.8%). Using combined positive score (CPS), the same authors found a PD-L1 expression with CPS ≥ 1 in 88 (67.2%) cases, including cases with low (1−19; n = 67, 51.2%), high (20−49; n = 13, 9.9%) and very high (≥50; n = 8, 6.1%) CPSs. 

#### 10.1.3. Tumor Microenvironment: Cytokines and Tumor Infiltrating Leucocytes (TILs)

In recent years, the prognostic and predictive role of tumor-infiltrating leucocytes (TILs) and cytokines levels of expression has been a topic of further interest. 

In a study [125], authors studied different populations of TILs in SNSCC and found out that level of CD8^+^ cell infiltration was a significant and independent favorable prognostic factor. However, high Foxp3+ Treg infiltration was also associated with favorable OS and DFS in SNSCC. 

In a series of SNCs [126], the authors analyzed different high-grade tumors. Among them, 16 were SNUCs, four SMARCB1-deficient sinonasal carcinomas, one SMARCA4-deficient carcinoma, five high-grade neuroendocrine carcinomas, one NC, one TCS, and two sinonasal N-ITAC. They focused on the expression of major histocompatibility complex molecules, the leukocyte infiltrates, and chemokines expression, finding that chemokines CXCL8 and CXCL5 were upregulated in high-grade sinonasal carcinomas, influencing leukocyte activation and trafficking, angiogenesis, metastasis, and cancer cells proliferation. On the other hand, several chemokines such as CCL28 and CCL14 were downregulated in SNUCs and high-grade neuroendocrine carcinomas compared with normal tissue. Targeting migration-related chemokines and their receptors in sinonasal tumors might be beneficial for immunotherapy.

A Spanish study [123] analyzed the tumor microenvironment immunotypes (TMIT), deriving from the combination of tumor-infiltrating lymphocyte density and PD-L1 expression, such as a biomarker for immunotherapy in 133 ITAC. The authors identified four immunotypes: type I (TIL^high^/PD-L1^pos^), type II (TIL^low^/PD-L1^neg^), type III (TIL^low^/PD-L1^pos^) and type IV (TIL^high^/PD-L1^neg^). They considered CD8^+^ cells as TILs. They found that intratumoral TILs are present in up to 65% of ITAC, while tumoral PD-L1 positivity was observed in 26% of cases. Furthermore, many TILs and TMIT types I and IV were associated with longer OS only. TMIT classification did not have additional prognostic value over TILs alone. Just 6% of cases were TMIT type I (CD8^high^/PD-L1^pos^), indicating that ITAC is a poorly immunogenic tumor type.

The Spanish [124] group also analyzed TMIT in a series of 57 SNSCCs. Approximately 88% of the cases displayed the presence of CD8^+^ TILs (19%—high; 69%—low) in the intratumoral compartment. From their analysis, 19% of cases were TMIT type I (CD8^high^/PD-L1^pos^). This result suggests that SNSCCs are immunogenic tumors, and that a subgroup might benefit from therapy with ICIs. This proportion is lower than in highly immunogenic tumors such as melanomas, renal cell and bladder cancers, HNSCCs or lung SCC (40–50% of the cases belong to TMIT I) [127], but still higher than in other SNCs, such as ITACs.

### 10.2. Clinical Data on the Efficacy of ICIs in Sinonasal Cancers

Most of the data on the potential efficacy of ICIs in various histological subtypes of sinonasal tumors come from case reports. Interestingly, the responses observed are not strictly related to PD-L1 expression, d-MMR phenotype, MSI or the presence of TILs.

A case report [128] presented two immunotherapy applications in SNCs and their relationship with other therapeutic strategies. The first patient was a 23-year-old man, treated with pembrolizumab in the second line for relapsing NC. After four cycles the patient underwent a partial response, but then a local progression of the disease was registered. He was offered hypofractionated stereotactic radiotherapy, and pembrolizumab was continued until a local complete response. The other patient was a 29-year-old man with a late local relapse of an SNSCC. Treatment with nivolumab and reirradiation was able to obtain a response, thus supporting the activity of this combination.

An Italian case report [129] described an impressive complete response to nivolumab in a 19-year-old man with metastatic SNUC. Interestingly, PD-L1 expression on tumor cells was 10% and no TILs were detected.

Hongo et al. [120] analyzed nine ICI-treated cases. They were characterized by maxillary sinus location, high clinical stage (IVa or IVb), negativity for high-risk HPV, and proficient MMR. Three out of nine (33.3%) patients obtained a response (two had a high score at TPS and CPS, while one of them had a score of 0).

In a retrospective analysis of SNSCC [130], five patients received ICIs as first-line therapy and six received ICIs as second- or beyond-line therapy. PD-L1 expression was observed in three cases (27.3%) with a median CPS of 0.2 (0–16). The median progression-free survival (PFS) was 4.2 months (95% CI, 0.3–8.1). Both PD-L1 status and treatment line (first line) showed a trend toward longer PFS. PFS and disease control rates at 6 months were 36.4% and 36.4%, respectively. The 6-month OS rate was 63.6%. Three patients achieved partial response (27.2%) with two responses lasting over 6 months. One responder had prior platinum and cetuximab therapy. Responses were observed regardless of PD-L1 expression (two responses in CPS 0 and one in CPS 16).

In limited and possibly biased case series, it is impossible to draw any conclusions about the clinical activity of ICIs in SNCs and prognostic or predictive markers. This should prompt the investigation of these treatments in specific cohorts of SNC with translational correlative studies.

To our knowledge, there is only one trial limited to SNC (specifically SNSCC) exploring the activity of pembrolizumab added to cisplatin and docetaxel chemotherapy, to increase the response rate to systemic induction therapy (NCT05027633). In other trials including ICI, SNCs are included with other histologies.

A phase II study with pembrolizumab and cetuximab is ongoing to treat R/M HNSCC, including SNSCC (NCT03082534). Another large phase II trial with nivolumab and ipilimumab is ongoing in patients with rare tumors, including SCC and adenocarcinoma of nasal and sinonasal sites (NCT02834013); similarly, in rare cancers, a phase II trial with nivolumab (AcSé trial) is ongoing, including SNCs (NCT03012581).

## 11. Conclusions

Few data exist on the efficacy and agnostic use of targeted agents in treating SNCs. Clinical trials performed so far with new molecules in R/M HNSCC have almost always excluded SNCs, thus leaving uncertainty about the transposability of the results in the setting of this rare cancer. There is a need for international collaboration to start clinical trials of these rare cancers, to investigate new drugs and combinations, leveraging tumor molecular profiles to define the most active therapeutic strategies.

## Figures and Tables

**Table 1 jcm-11-06787-t001:** SNSCC’s potentially actionable mutations.

Gene	Findings	References	Types of Alterations	Variant Classification of Alterations	Target Drugs (Approved at Least by One Regulatory Agency in Other Cancer Settings)	Principal Treatment Indications
EGFR	≈14% de novo SNSCCC (9/63) ≈77% of ISP-associated SNSCC (17/22)	Udager et al. 2015 [9] Sasaki et al. 2018 [10]	Exon 20ins	Insertion In frame	Amivantamab Mobocertinib	NSCLC
ERBB2	21% (8/38)	Lòpez et al. 2011 [15]	CNG	Amplification	Trastuzumab, Lapatinib, Pertuzumab, Ado-trastuzumab emtansine, Fam-trastuzumab deruxtecan, Margetuximab, Neratinib	Breast cancer, gastric cancer

**Table 2 jcm-11-06787-t002:** ITAC’s potentially actionable mutations. CNG = copy number gain.

Gene	Findings	References	Types of Alterations	Variant Classification of Alterations	Target Drugs (Approved at Least by One Regulatory Agency in Other Cancer Settings)	Principal Treatment Indication (Tumors for Which Are Approved)
MET	0–64% (46/72)	Projetti et al. 2015. [48]	20 CNG	Amplification	Capmatinib	NSCLC
			(S156L)	(Missense)	(No drugs)	
EGFR	2–63% (27/43)	Szablewski et al., 2013 [54]	5 CNG	Amplification	Afatinib	NSCLC
PIK3CA	10–22% (5/48; 11/50)	Sánchez-Fernández et al., 2021 [5]; Riobello et al., 2021 [53]	Q546R, H1047R, K111E	Missense	Alpelisib	Breast cancer
			(D939G), (E726K), (V1534M), (D454G)	(Missense)	(No drugs)	
NRAS	8% (4/48)	Sánchez-Fernández et al., 2021 [5]	G12T	Missense	Bimetinib	Melanoma
			(6-10CNG)	(Amplification)	(No drugs)	
BRCA 1 and BRCA 2	8–14% (4/48) (7/50)	Sánchez-Fernández et al., 2021 [5]. Riobello et al., 2021 [53]	R1347G	Missense	Olaparib, talazoparib, niraparib	Breast cancer, ovarian cancer, prostatic cancer
			L3326*, K3226*	Nonsense		
			5 CNG	Amplification		
			(P1603Rfs*13), (Q1111Nfs*5)	(Frameshift)	(No drugs)	
			(V1534M)	(Missense)	(No drugs)	
ATM	8–16% (4/48) (8/50)	Sánchez-Fernández et al., 2021 [5]. Riobello et al., 2021 [53]	Q684P, P1054R, D1853V, V410A	Missense	Olaparib	Prostatic cancer
AR	0–20% (10/50)	Riobello et al., 2021 [53]	Q79–Q80 dupl.,	Inframe duplication	Bicalutamide, leuprolina	Salivary glands cancer
			(Q77–Q80 del.), (Q79–Q80 del.,)	(Deletion)	(No drugs)	
ERBB2	0–6% (3/50)	Riobello et al., 2021 [53]	S310F	Missense	Trastuzumab, Pertuzumab, Ado-trastuzumab emtansine, Fam-trastuzumab deruxtecan, Margetuximab, Neratinib, Lapatinib	Breast cancer, Gastric cancer
BRAF	0–6% (1/18)	Franchi et al., 2014 [40]	V600E	Missense	Dabrafenib, cobimetinib +MEK inhibitors	Melanoma
			(D594N)	(Missense)	(No drugs)	
IDH1	sporadic	Riobello et al., 2021 [53]	R132C	Missense	Ivosidenib	Ductal bile carcinoma

**Table 3 jcm-11-06787-t003:** N-ITAC’s potentially actionable mutations.

Gene	Findings	References	Type of Alterations	Variant Classification of Alterations	Target Drugs (Approved at Least by One Regulatory Agency in Other Cancer Settings)	Principal Treatment Indications
BRAF	16% (2/12)	Franchi et al., 2013 [63]	V600E	Missense	Dabrafenib + trametinib	Melanoma
NTRK3	Case report (1 patient.)	Andreasen et al., 2017 [64]	ETV6-NTRK3 fusions	Translocation	Entrectenib, larotrectinib	All NTRK rearranged cancers
RET	Case report (1 patient.)	Andreasen et al., 2018 [65]	ETV6-RET fusions	Translocation	Selpercatinib, pralsetinib	All RET rearranged cancers

**Table 4 jcm-11-06787-t004:** SNEC’s potentially actionable mutations.

Gene	Findings	References	Types of Alterations	Variant Classification of Alterations	Target Drugs (Approved at Least by One Regulatory Agency in Other Cancer Settings)	Principal Treatment Indication (Tumors for Which Are Approved)
SMARCB1	14% (2/14)	Libera et al. 2021. [77]		Deletions, Nonsense	Tazemetostat	Epithelioid Sarcoma
IDH2	11–83% (1/9; 8/39; 5/6) *	Gloss et al. 2021. [70] Riobello et al. 2019 [69]. Dogan et al. 2019. [71]	R172S, R172G, R172M, (R127T), (CNG).	Insertions, deletions	Enasidenib	Acute myeloid leukemia

* Evaluated as SNEC in a cohort of IDH2-mutated sinonasal tumor samples (as numerator number of SNEC and as denominator total number of samples).

**Table 5 jcm-11-06787-t005:** SNUC’s potentially actionable mutations.

Gene	Findings (%)	References	Types of Alterations	Variant Classification of Alterations	Target Drugs (Approved at Least by One Regulatory Agency in Other Cancer Settings)	Principal Treatment Indication (Tumors for Which Are Approved)
IDH2	31–88% (11/36, 14/16)	Dogan et al. 2019, Riobello et al. 2019 [69,71]	R172S, R172G, R172T, R172M, copy number gains	Insertions Deletions	Enasidenib	Acute myeloid leukemia
SMARCB1/INI 1	43% (6/14), case report (1 patient)	Chitguppi et al. 2020 [90], Saleh et al. 2022 [91]		Deletions	Tazemetostat	Epithelioid Sarcoma
ERBB2	Highly	Takahashi et al. 2016 [94]		Overexpression (by Western blot)	Trastuzumab-emtansine, Trastuzumab-deruxtecan, Trastuzumab, pertuzumab, lapatinib, margetuximab, Neratinib,	Breast cancer, Adenocarcinoma of gastroesophageal junction
EGFR	27% (3/11)	Chernock et al. 2009 [92]		Overexpression	Cetuximab, Panitumumab	NSCLC, colorectal cancer, Head and neck SCC

**Table 6 jcm-11-06787-t006:** Immuno-markers in SNC.

Immune-Markers in SNC	Findings	References
d-MMR/MSI	2% ITAC 2–21% SNSCC	Martínez et al. 2009. Uryu et al. 2006, Hongo et al. 2021, Hermsen et al. 2009 [118,119,120,121]
PD-L1 expression by IHC > 5%	34% (18/53) SNSCC 17% (22/126) ITAC	Riobello et al. 2018 [122]
PD-L1 expression by IHC > 50%	26% (14/53) SNSCC 3% (4/126) ITAC	Riobello et al. 2018 [122]
PD-L1 expression by CPS ≥ 1	67.2% (88/131) SNSCC	Hongo et al. 2021 [120]
CD8^high^/PD-L1^pos^	6% (8/133) ITAC 19% (11/57) SNSCC	Garca-Marín et al. 2020 and 2021 [123,124]

## Data Availability

Not applicable.

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
