# Peer review of "Molecular Basis and Rationale for the Use of Targeted Agents and Immunotherapy in Sinonasal Cancers"

_jcm, 2022, doi:10.3390/jcm11226787_

Round 1

Reviewer 1 Report

Esposito et al conducted a very thorough and detailed review on the molecular profiles of sinonasal cancers. I’ld like to congratulate them for this in-depth and timely work on such an orphan group of diseases.  

Major comments:

-       In my opinion, the current manuscript should be shorten in length. While the information provided for each of the histologic subtypes is very detailed, it is not presented in a succinct manner. The authors list and describe the findings of each study in the text not always in an ordered/consistent fashion, and it is hard to follow at times.  Since there are  tables listing the molecular alterations,  the text in each section should summarize the most relevant information and focus on the clinical implications of the findings on a most generalized way.  

-       Methodology used to conduct the search should be better described (criteria for selection if there were any e.g. number of cases included, etc) .

-       The sections do not seem to follow a logical order at times and should be addressed. Two examples: IPs are in a different section while they were previously described within  the SNSCC section.  OPs are actually included  in the same section which is inconsistent. Similarly, immuno-markers subsection is at the same level of PD-L1, tumor microenvironment. are they are not all immune markers?

-       I would suggest to include a table to give a summary of the immune biomarkers portion 

-       I would consider add a section or a table of the current targeted therapies/immunotherapies being tested in clinical trials for SNCs specifically or that included them as this would be useful for the reader and timely for a manuscript like this. 

-       Summary tables should include the N of patients for each study as would facilitate interpretation of the results (not the same 50% of 10 samples, than 150 samples)  

-       The manuscript would benefit from an English writing revision. 

Other  comments

- Make sure the referencing is consistent and that all statements are supported/backed up by literature.

- Line 15-18: Suggest which are the specific improvements in neoadjuvant/palliative chemotherapy that have led to survival improvements? 

-  Line 3: I believe the word malignant seems out of context. 

- Line 10-12: suggest to revise sentence and put the subject (these tumors) early on for better comprehension. 

- Paragraph 4: If targeted therapies and immunotherapies are not approved ad there is no evidence, they should not in fact be administered in the curative setting nor palliative unless it's within clinical trials or expanded usage based on biomarker (e.g. :Pd-L1, TMB) - I'ld suggest to clarify this.  Regarding targeted therapies: the statement on "potentially possess alterations in different pathways" is a bit broad, should be backed up by references. 

- Substitute HER-2 activity for Efficacy of antiHER-2 agents shall be explored ...

- Please review and rephrase this sentence "In detail, studies about the genetic characterization of SNSCC showed that the IP-associated carcinomas have some molecular peculiarity". Why in detail and what are molecular peculiarities? 

- Please add an explanation along this statement: However, there is only a certain degree of overlay between the genetic mutation’s panorama of these two tumor types   

- ITAC histological subgroups are mentioned but not described and somehow the reader must be lost as the immediate prior comparison was between colon and ITAC

- Poor prognosis and worse survival seems redundant. 

- Immunotherapy section: This section is focused exclusively on ICI - I'ld suggest to change the name accordingly.  

- Conclusions: I suggest the authors to tone down the following statement " we suggest that all SNC patients should undergo genomic sequencing evaluation, in order to find potentially actionable targets". I don't think the current evidence supports to test all SNCs if not within a clinical trial or research program to allow expanded usage of targeted therapies, especially when we still don't have much evidence of efficacy per se. Also, consider narrow it down to specific settings (R/M) or specific histologies ? 

Author Response

Paolo Bossi

Medical oncology Unit, University of Brescia, ASST-Spedali Civili Brescia, Italy

[email protected]

Dr Rainie Zhang
Section Managing Editor
MDPI JCM Editorial Office
St. Alban-Anlage 66, 4052 Basel, Switzerland
E-Mail: [email protected]
https://www.mdpi.com/journal/jcm                                                                                              November, 3, 2022

Subject:  Manuscript ID: jcm-1996263

Dear Dr Rainie Zhang and dear reviewers,

Thank you for your letter and for the opportunity to revise our paper “Molecular Basis and Rationale for The Use Of Targeted Agents And Immunotherapy In Sinonasal Cancers”.

The suggestions offered by the reviewers have been very helpful and have improved the quality of the manuscript.

I have included the reviewer comments immediately after this letter and responded to them individually, indicating exactly how we addressed each concern or problem and describing the changes we have made. The revisions have been approved by all authors. We decided to send two different versions of the manuscript, one with track changes and one named “clean version”, that can be considered for further revisions.

We tried to re-organize and shorten in length the manuscript. The manuscript has undergone an extensive English revision, as requested.

We hope the revised manuscript will better suit the  Journal of Clinical Medicine but are happy to consider further revisions, and we thank you for your continued interest in our research.

Sincerely,

Paolo Bossi,   Medical oncology Unit, University of Brescia, ASST-Spedali Civili Brescia, Italy

Reviewer Comments, Author Responses and Manuscript Changes

Reviewer one:

Major comments:

  • In my opinion, the current manuscript should be shorten in length. While the information provided for each of the histologic subtypes is very detailed, it is not presented in a succinct manner. The authors list and describe the findings of each study in the text not always in an ordered/consistent fashion, and it is hard to follow at times.  Since there are  tables listing the molecular alterations,  the text in each section should summarize the most relevant information and focus on the clinical implications of the findings on a most generalized way.  

Response: Thanks for this suggestion. In the new version, we shortened the manuscript, mainly by deleting non useful parts (in particular those on non-actionable gene alterations).

  • Methodology used to conduct the search should be better described (criteria for selection if there were any e.g. number of cases included, etc) .

Response: Studies were identified with searches using PubMed (July 2022). Articles concerning sinonasal epithelial cancer and molecular alterations written in English were included. We added more details on keywords and methods of research in the Introduction section.

  • The sections do not seem to follow a logical order at times and should be addressed. Two examples: IPs are in a different section while they were previously described within  the SNSCC section.  OPs are actually included  in the same section which is inconsistent. Similarly, immuno-markers subsection is at the same level of PD-L1, tumor microenvironment. are they are not all immune markers?

Response: Thanks for your suggestion. We reviewed the organization of ICI section: 9.1 section is about immunomarkers (we introduced subsections 9.1.1, 9.1.2 and 9.1.3 to better depict them), while clinical evidence of ICI can be considered as 9.2 section. For the IP and OP section, we re-arranged the chapter by merging the parts on IP-associated SNSCC with the aim of making it more cohesive and fluid to read.

  • I would suggest to include a table to give a summary of the immune biomarkers portion 

Response: we have accepted the suggestion, as this could improve the manuscript. The table has been inserted in the text as Table nr 6 and it is reported below

Immune-markers in SNC

Findings

References

d-MMR/MSI

2% in ITAC

2-21% in SNSCC

Martinez et al. 2009.

Uryu et al. 2006, Hongo et al. 2021, Hermsen et al. 2009

PD-L1 expression by IHC > 5%

34% (18/53) in SNSCC

17% (22/126) in ITAC

Riobello et al. 2018

PD-L1 expression by IHC > 50%

26% (14/53) in SNSCC

3% (4/126) in ITAC

Riobello et al. 2018

PD-L1 expression by CPS ≥ 1

67.2% (88/131) in SNSCC

Hongo et al. 2021

CD8high/PD-L1pos

6% (8/133) ITAC

19% (11/57) SNSCC

Garcia-Marin et al 2020 and 2021

Table 6: Immunomarkers in SNCs.

  • I would consider add a section or a table of the current targeted therapies/immunotherapies being tested in clinical trials for SNCs specifically or that included them as this would be useful for the reader and timely for a manuscript like this. 

Response: thank you for your comment. We discussed about adding a specific section on ongoing trials for SNCs. Due to the paucity of studies on these topics, we have preferred not to devote a particular section to trials, but to leave them in the individual sections (e.g., in the final part of the chapter on ICIs there is a quick roundup of the ongoing studies on ICIs in CNS).

  • Summary tables should include the N of patients for each study as would facilitate interpretation of the results (not the same 50% of 10 samples, than 150 samples)  

Response: we have supplemented with absolute numbers. We only point out that in the section on SNEC, in the studies concerning the IDH2 mutation, there were no absolute numbers of how many SNEC had that mutation, but it was explained that out of a total number of tumors carrying that mutation, some of them were considered SNEC. We added a specification below SNEC table.

. 7)    The manuscript would benefit from an English writing revision. 

Response: the manuscript underwent an English revision

Other  comments

- Make sure the referencing is consistent and that all statements are supported/backed up by literature.

Thank you for your advice.

- Line 15-18: Suggest which are the specific improvements in neoadjuvant/palliative chemotherapy that have led to survival improvements? 

The sentence can be modified in “Advances in imaging techniques, surgical and endoscopic approach, radiotherapy modalities (intensity-modulated radiation therapy, volumetric modulated arc therapy and heavy-ion radiations) and neoadjuvant chemotherapy-containing strategies have shown promising results in improving outcomes.” Data on neoadjuvant chemotherapy-containing approaches can be derived from SINTART studies.

-  Line 3: I believe the word malignant seems out of context. 

Yes, it is. We deleted the word “malignant”

- Line 10-12: suggest to revise sentence and put the subject (these tumors) early on for better comprehension. 

The sentence has been changed in “There are difficulties in diagnosis and treatment of these tumors due to rarity, histological diversity and proximity to vital structures (such as orbit, skull base, and brain).”

- Paragraph 4: If targeted therapies and immunotherapies are not approved ad there is no evidence, they should not in fact be administered in the curative setting nor palliative unless it's within clinical trials or expanded usage based on biomarker (e.g. :Pd-L1, TMB) - I'ld suggest to clarify this.  Regarding targeted therapies: the statement on "potentially possess alterations in different pathways" is a bit broad, should be backed up by references. 

We agree with your suggestions and we have edited the paragraph.

- Substitute HER-2 activity for Efficacy of antiHER-2 agents shall be explored ...

We agree with your suggestion and we have edited as requested.

- Please review and rephrase this sentenced "In detail, studies about the genetic characterization of SNSCC showed that the IP-associated carcinomas have some molecular peculiarity". Why in detail and what are molecular peculiarities? 

We rearranged all the SNSCC chapter and eliminated the sentence .

- Please add an explanation along this statement: However, there is only a certain degree of overlay between the genetic mutation’s panorama of these two tumor types   

KRAS, HRAS, and NRAS mutations and EGFR copy number gains are less frequent in ITAC than in colorectal cancer. We rearranged the ITAC’s section.

- ITAC histological subgroups are mentioned but not described and somehow the reader must be lost as the immediate prior comparison was between colon and ITAC

The sentence about ITAC subgroups can be deleted.  We edited the ITAC section.

- Poor prognosis and worse survival seems redundant. 

We agree with your suggestion

- Immunotherapy section: This section is focused exclusively on ICI - I'ld suggest to change the name accordingly.  

We agree and we changed the title of the section.

- Conclusions: I suggest the authors to tone down the following statement " we suggest that all SNC patients should undergo genomic sequencing evaluation, in order to find potentially actionable targets". I don't think the current evidence supports to test all SNCs if not within a clinical trial or research program to allow expanded usage of targeted therapies, especially when we still don't have much evidence of efficacy per se. Also, consider narrow it down to specific settings (R/M) or specific histologies ? 

Response: Thank you for your suggestion. We decided to delete the sentence. We cannot conclude if all SNC could undergo NGS. In R/M SNCs, an agnostic approach could be offered, but we need more studies on the efficacy of target therapies/ICI in SNCs.

Reviewer 2 Report

This is a comprehensive well written very organized review of the existing literature on molecular pathways, genetic mutations and immunology of sinonasal epithelial malignancies as they relate to potential therapeutic implications, specifically use of targeted agents and immunotherapy. This is a valuable resource for any clinician interested in treating sinonasal cancer.

For the sake of completeness, I suggest to include a paragraph highlighting the emerging role of HPV in sinonasal cancer and the potential molecular / genomic signatures that may have therapeutic implications in the future.

Minor typographical errors:

Page 1: In Italy malignant more than 300 SNC are registered every year 

Page 5: During last years, 

Author Response

Comment 1: For the sake of completeness, I suggest to include a paragraph highlighting the emerging role of HPV in sinonasal cancer and the potential molecular / genomic signatures that may have therapeutic implications in the future.

Response: thank you. We decided to highlight the emerging role of HPV in sinonasal cancer in the SNSCC’s section.

Minor typographical errors:

Page 1: In Italy malignant more than 300 SNC are registered every year 

  • The word “malignant” was deleted.

Page 5: During last years, 

  • We edited as requested.

Round 2

Reviewer 1 Report

Dear authors, 

Congratulations and thanks for the work done on addressing the suggestions and comments provided.

Regarding the following statement in the conclusions section: "we suggest that all SNC patients should undergo genomic sequencing evaluation, in order to find potentially actionable targets"  the response letter states that the sentence has been removed but I can still see it in the submitted reviewed manuscript. Please make sure this is addressed as discussed.